# Proxies for use in biochar decay models: Hydropyrolysis, electric conductivity, and H/C$_{org}$ molar ratio

Nikolas Hagemann[1,2,3*], Hans-Peter Schmidt[2], Thomas D. Bucheli[1],
Jannis Grafmüller[3,4], Silvio Vosswinkel[5,6], Volker Herdegen[5], William Meredith[7],
Clement N. Uguna[7], Colin E. Snape[7]

**1** Environmental Analytics, Agroscope, Zurich, Switzerland, **2** Ithaka Institute, Arbaz, Switzerland, **3** Ithaka Institute, Goldbach, Germany, **4** Institute for Sustainable Energy Systems, Offenburg University of Applied Sciences, Offenburg, Germany, **5** Institute of Thermal-, Environmental- and Resources' Process Engineering, Technische Universität Bergakademie Freiberg, Freiberg, Germany, **6** Eurofins Umwelt-Ost GmbH, Bobritzsch-Hilbersdorf, Germany, **7** Faculty of Engineering, University of Nottingham, Nottingham, United Kingdom

\* hagemann@ithaka-institut.org, nikolas.hagemann@agroscope.admin.ch

## Abstract

Biochar is a carbon-rich material produced via pyrolysis that is increasingly recognized for its role in carbon sequestration, particularly through its application in agriculture and materials. However, accurately predicting the long-term persistence of biochar in the environment remains challenging. While incubation trials have been widely used to assess biochar degradation, their extrapolation beyond centennial timescales is uncertain. In this study, we evaluate the consistency between three physicochemical characterization methods that are considered as proxies for biochar persistence—hydropyrolysis (HyPy), solid-state electric conductivity (SEC), and elemental analysis to obtain molar hydrogen:carbon ratios. We produced 42 biochars from straw and wood using a continuously operated pilot-scale auger reactor at temperatures ranging from 400 to 800 °C under otherwise constant pyrolysis conditions. We then systematically analyzed the elemental composition, SEC and the fraction of biochar carbon that is resistant to HyPy (BC$_{HyPy}$). Hydropyrolysis eliminates all free and covalently bound non-aromatic species and all aromatic species consisting of up to seven fused rings. Our results confirm that BC$_{HyPy}$ content increases with pyrolysis temperature and stabilizes above 600–680 °C, reaching >90% of total carbon in high-temperature biochars. Similarly, SEC increased exponentially with pyrolysis severity, correlating strongly with BC$_{HyPy}$ and H/C molar ratio. The latter has so far been used to predict biochar persistence. Our findings from a controlled temperature series of biochars highlight that SEC and BC$_{HyPy}$ could be useful proxies for parameterizing multi-pool decay models of biochars produced in practice.

**Data availability statement:** All relevant data are within the manuscript and its Supporting Information files.

**Funding:** The research at the University of Nottingham was supported by: (i) the Biotechnology and Biological Sciences Research Council [BBSRC, the Biochar Demonstrator, grant number BB/V011596/1] as part of the UKRI Greenhouse Gas Removal programme and (ii) the Department of Energy Security and Net Zero (DESNZ) through the Direct Air Capture and Greenhouse Gas Removal Programme Phases 1 and 2 for the grant "Bio-waste to Biochar (B to B) via Hydrothermal Carbonisation and PostCarbonisation". The research at Ithaka was funded by the Deutsche Forschungsgemeinschaft (DFG, German Research Foundation) – 467391808. The funders had no role in study design, data collection and analysis, decision to publish, or preparation of the manuscript.

**Competing interests:** I have read the journal's policy and the authors of this manuscript have the following competing interests: Hans-Peter Schmidt reports a relationship with Carbon Standards AG that includes: board membership. Hans-Peter Schmidt and Nikolas Hagemann are authors of the Carbon Standards AG guidelines for the certification of biochar quality assurance (EBC) and biochar-based carbon sinks (EBC C-Sink / Global Biochar C-Sink, Global Artisan C-Sink). All other authors have declared that no competing interests exist. There are no patents or products related to the submission at this time. The competing interests do not alter my adherence to PLOS ONE policies of sharing data and materials.

## 1. Introduction

Biochar is a pyrogenic carbonaceous material that is deliberately produced by biomass pyrolysis and used in a non-oxidative manner. To achieve carbon sequestration (pyrogenic carbon capture and storage, PyCCS) [1], biochar is applied as an additive in materials like concrete, asphalt, and composites or in agriculture, e.g., as a manure additive or as a nutrient carrier in slow-release fertilizers. These agricultural applications ultimately introduce biochar into soil, where the biochar-carbon is stored, which is a key aspect of PyCCS. Soil-applied biochar has been third-party certified as carbon dioxide removal (CDR) since 2020 [2–4]. To account for the climate effect of biochar-CDR, it is crucial to determine the quantity of biochar-carbon that stays sequestered at any point in time after its soil application. Therefore, it is necessary to predict the persistence of biochar in soil.

The incubation of biochar in soil or similar matrices combined with quantifying $CO_2$ release is an intuitive and widespread approach to assessing biochar degradation. However, as shown by Azzi *et al.* (2024) and Sanei *et al.* (2025) [5,6], biochar incubation studies that stretch at maximum over several years are not suitable to extrapolate biochar degradation beyond decades to centuries. Instead, multi-pool decay models based on the combination of data obtained from incubation studies, accelerated aging experiments, and physicochemical characterization are suggested, which includes spectroscopy, chromatography, microscopy, and elemental analysis to calculate the hydrogen/carbon (H/C) molar ratio [6]. Here, we propose the quantification of biochar-carbon resisting hydropyrolysis ($BC_{Hypy}$) and the solid-state electric conductivity (SEC) of biochar as two novel physicochemical characterization methods to support the parametrization of such novel decay models for individual biochars.

Hydropyrolysis (HyPy) [7], a pyrolysis process conducted at 550 °C under high-pressure hydrogen (150 bar), is an analytical technique used to remove thermally labile carbon compounds. Gas chromatography-mass spectrometry (GC-MS) analysis of the volatilized compounds suggests that HyPy predominantly releases species with fewer than eight condensed aromatic rings, though this observation may be constrained by the volatility and ionization characteristics required for GC-MS analysis [8]. The volatilized compounds included coronene, which is a molecule consisting of 7 fused benzene rings and 24 carbon atoms [9]. The carbon that is not volatilized under HyPy is referred to as $BC_{HyPy}$ (i.e., black carbon after HyPy) or SPAC (stable polycyclic aromatic carbon), which is operationally defined as highly condensed carbon (>7 aromatic rings). This method was originally developed to remove sorbed organic carbon from historic charcoal samples and to thus avoid a "dilution" of the radiocarbon signature of the original pyrogenic carbon [10]. Process parameters were selected to avoid hydrogasification, i.e., the conversion of carbonaceous compounds into methane, and the formation of secondary char [7,11].

Hydropyrolysis has been used to assess the thermal stability and composition of biochars since 2015 [12]. The thermally labile fraction of biochars, i.e., non-$BC_{HyPy}$ including compounds with up to seven condensed aromatic rings, is considered to be more susceptible to (microbial) degradation than $BC_{HyPy}$, for which centennial

persistence was postulated [13–15]. The $BC_{HyPy}$ content increases with pyrolysis severity and there is a strong inverse correlation between the increase in $BC_{HyPy}$ content, with minimal change below 450 °C, rapid growth between 500–700 °C and a plateau at higher temperatures [12].

Solid-state electric conductivity arises from biochar's aromatic carbon structure, where conjugated π-electrons enable electron transfer [16]. Higher pyrolysis temperatures enhance this effect by increasing graphitic ordering and reducing resistivity [17]. To quantify SEC, a two-probe packed-bed technique is used, where electrical resistance is measured under applied compressive pressure, and conductivity is calculated based on resistivity and bed length [17–19]. While higher SEC has already been correlated with increased contaminant remediation [18], we here aim to test SEC as a proxy for biochar persistence due to its direct link with carbon speciation.

To test and eventually establish these two analytical parameters for biochar persistence evaluation, we systematically produced 42 biochars from wood (the most common feedstock used at industrial scale in Europe [20]) and straw (a common crop residue used in research studies with a lower lignin and higher ash content [21]) at increasing temperatures between 400–800 °C. We then tested the consistency between these two methods, but also with the H/C molar ratio, which has previously been used as a proxy to estimate biochar persistence due to its link to the pyrolysis temperature and hence to biochar persistence.

Unlike previous studies, biochars were not produced in lab-based batch pyrolysis setups (e.g., thermo-gravimetric analysis, muffle furnace), but in a continuously operating auger pyrolysis reactor at a pilot scale of 1 kg biomass input per hour [22]. This setup provides conditions that are similar to most commercial pyrolysis units while allowing the application of a wide range of well-defined pyrolysis conditions, such as temperature and residence time. We aimed for a correlation of $BC_{HyPy}$, SEC, pyrolysis temperature, and H/C molar ratio and to discuss the suitability of these parameters for the parametrization of novel decay models for industrial biochars dedicated to PyCCS.

## 2. Materials and methods

### 2.1 Biochar production

Pellets were produced with a diameter of 6 mm on a roller wheel mill (WK230, EverTec, Groß-Zimmern, Germany) from straw (Jumbo/Coop, Basel, Switzerland, 5.9% ash content, cf. S1 Table) and softwood (Allspan Spanverarbeitung GmbH, Karlsruhe, Germany, 0.4% ash content), respectively. Three batches each of straw and wood pellets were produced at different points in time from the same biomass (first batch: biochars produced at 400–600 °C in 50°-steps, second batch: 620–800 °C in 20°-steps, third batch: replicate biochars produced at 600 and 700 °C). Smaller temperature increments starting at 600 °C were chosen, as this is the temperature range of most commercial pyrolysis plants and the greatest changes in carbon speciation were expected [23]. The biomass composition is shown in S1 Table. Experimental pyrolysis was performed with a PYREKA research pyrolysis unit (Pyreg GmbH, Dörth, Germany), a continuously operated auger reactor [22] adjusted to a residence time of 10 min. This setup is described in detail in Hagemann et al. [22]. Feeding rate was kept constant for each batch of feedstock and was in the range 0.4–0.7 kg h$^{-1}$. The reactor was purged with 2 L min$^{-1}$ $N_2$. Biochars were collected for 30–45 min to achieve 50–150 g per sample. After changing the pyrolysis temperature during continuous feeding of biomass into the reactor, biochar produced during the subsequent 30 min was discarded; when the input of a new biomass was started, the production of the first 45 min was discarded. We labeled the biochars beginning with the feedstock (i.e., W for wood, S for straw) followed by the pyrolysis temperature (e.g., a biochar produced at 600 °C from straw pellets is denoted as S600). If the temperature indication is followed by the capital letters A-C (e.g., W600A), it is a replicate of the biochar production (intra-day precision). The replicates were produced during an ongoing continuous pyrolysis, each with a sampling interval of 30 min.

### 2.2 Biochar characterization

Elemental analysis (CHN) was performed according to DIN 51732. The SEC was determined while the ground biochar (< 0.2 mm) was subjected to a pressure of 10 kN between two electrodes of the "Black Gauß I" device, which equals 30 MPa.

The apparatus and procedure are described in detail elsewhere [19]. Both methods are compliant with the analytical guidelines of the European Biochar Certificate [24]. The ash content needed to express the $BC_{HyPy}$ content on a dry and ash-free (daf) basis was quantified according to DIN 51719 (550 °C). Oxygen (O) was determined on a vario EL-cube (elementar, Langenselbold, Germany).

For HyPy, 100−200 mg of biochar sample were loaded with a Mo catalyst using an aqueous/methanol (80%/20%) 0.2 M solution of ammonium dioxydithiomolybdate [$(NH_4)_2MoO_2S_2$]. Catalyst weight was ~10% of the sample weight. The catalyst-loaded biochar was dried (110 °C, 24 h) and samples were placed in quartz tubes (20 mm long), sealed with a sintered disc at the base, and placed in the HyPy reactor. The samples were heated at a rate of 300 °C min$^{-1}$ from 50 to 250 °C (i.e., within 40 seconds), then heated at 8 °C min$^{-1}$ from 250 °C until the final temperature of 550 °C, which was held then for 2 min under a hydrogen pressure of 15 MPa. More details are described elsewhere [11]. A controlled constant hydrogen sweep-gas flow of 5 L min$^{-1}$ in the reactor, measured at ambient temperature and pressure, ensured that the labile products were quickly removed from the samples. The mass and carbon content of the HyPy residue were quantified.

## 3. Results

Carbon contents of the woody biochars produced at 400–800 °C were between 81–92% and increased with increasing temperature (S2 Table) while H/C molar ratios decreased from 0.43 to 0.10. Straw biochars had carbon contents of 62–70% with a peak at 550 °C, despite a continuously decreasing H/C molar ratio from 0.48 to 0.16 for 400–800 °C.

Solid-state electric conductivity increased exponentially with increasing pyrolysis temperature and ranged from $10^{-5}$ to $10^3$ mS cm$^{-1}$ (Fig 1a). Notably, the straw biochars had consistently higher conductivity than woody biochars for the range of 400–700 °C, despite having lower carbon and lower $BC_{HyPy}$ contents, indicating that other factors than pyrolysis temperature also affect conductivity.

Hydropyrolysis revealed a $BC_{HyPy}$ fraction of 46.7% and 59.0% of total carbon (TC) for straw and wood biochar produced at 400 °C, respectively. When pyrolyzed at 680 °C or above, $BC_{HyPy}$ fraction of biochar from both feedstocks was > 90%; for woody biochars, this value was already achieved at 600 °C (S2 Table, Fig 1c). For temperatures above 600 °C, $BC_{HyPy}$ content stabilized (varied) between 90–99% and 90–97% for woody and straw biochars, respectively (S2 Table, Fig 1c).

Both the SEC and the content of $BC_{HyPy}$ increased with decreasing H/C molar ratio (Fig 1b and 1d). Whereas the content of $BC_{HyPy}$ of straw and wood biochars was almost identical at a given H/C ratio, SEC was systematically higher for straw biochar, which consistently had higher molar H/C ratios for pyrolysis temperatures above 680 °C (Fig 1e). Woody biochar reached 90% $BC_{HyPy}$ when SEC was above 0.1 mS cm$^{-1}$, and H/C below 0.27. Straw biochar reached 90% $BC_{HyPy}$ only when SEC was above 26 mS cm$^{-1}$ and the H/C below 0.20. Triplicate biochar production and analysis showed excellent repeatabilities for both feedstocks. Values of $BC_{HyPy}$ varied by approximately 1%, which is lower than the variation coefficient for H/C molar ratios, while the variation coefficient was 7–23% for SEC (Table 1). It should be noted that in our dataset, SEC spans across eight orders of magnitude ($10^{-5}$–$10^3$ mS cm$^{-1}$), while $BC_{HyPy}$ and H/C molar ratio vary within the same order of magnitude across all biochars produced at 400–800 °C. Therefore, SEC is more sensitive to small changes in the carbon speciation within our biochar production replicates, as the variation coefficient of replicate measurements was 2.1% (Table 1).

## 4. Discussion

### 4.1 Biochar properties determined by feedstock type, preparation, and pyrolysis conditions

Pyrolysis leads to the volatilization of low molecular weight carbonaceous compounds rich in O and H, resulting in an increase in carbon content and a decrease in H/C molar ratio of the solid product [12,21], which was confirmed in the present study. However, when using straw, a high-ash biomass (5.9%, S1 Table), the carbon content of the resulting

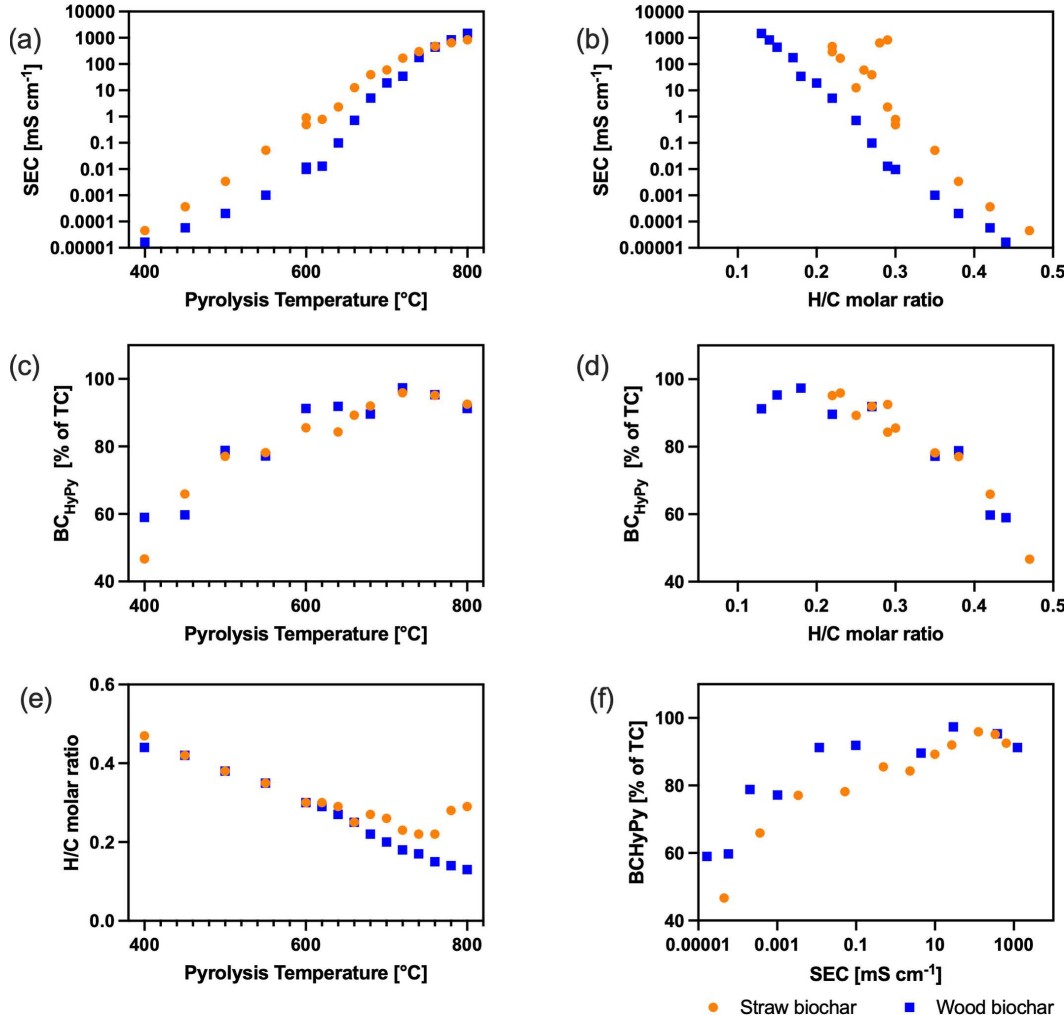

**Fig 1. Biochar properties.** Solid-state electric conductivity (SEC, mS cm⁻¹, a, b) and BC$_{HyPy}$ content in percent of total carbon (TC) of biochars (c, d, f) produced at defined temperatures in the range of 400-800 °C from straw and wood pellets. The pyrolysis temperature (a, c, e) and biochar H/C molar ratios (b, d) are used as parameters. Biochar H/C molar ratio is presented as a function of pyrolysis temperature (e). Raw data is presented in S1 Table.

biochars decreased at temperatures above 550 °C because of the non-proportional accumulation of mineral matter [12], which was not observed for the biochars made from low-ash wood (0.4%).

While low-temperature biochar is an electrical insulator, high-temperature biochar is electrically conductive due to the presence of conjugated π-electrons in its aromatic carbon structure. The degree of graphitization and aromaticity strongly influence conductivity, as a higher proportion of sp²-hybridized carbon enhances charge transport. Biochars produced at higher pyrolysis temperatures exhibit increased conductivity due to larger graphitic ordering and π-electron delocalization as the result of a higher degree of polycondensation [25,26]. X-ray diffraction and ¹³C nuclear magnetic resonance spectroscopy revealed that aromaticity and the degree of polycondensation increase with pyrolysis temperature, which is further indicated by the reduction of the H/C molar ratio [21,23,27–29]. The data on the SEC presented here fit well into the state of knowledge: at higher pyrolysis temperatures, biochars present a higher degree of polycondensation, which results in higher electrical conductivity. It should be emphasized that the increase occurred exponentially. In the parameter range investigated here (up to 800 °C and H/C = 0.1), a linear increase in the logarithmic SEC can be observed without any signs

**Table 1. Properties of biochars produced in triplicates.**

| | TC (%) | H (%) | H/C ratio | | BC$_{HyPy}$ (% of TC) | | SEC (mS cm$^{-1}$) | |
|---|---|---|---|---|---|---|---|---|
| | | | | Var. Coeff | | Var. Coeff | | Var. Coeff |
| W600A | 88.7 | 2.6 | 0.36 | 1.3% | 91.1 | 1.0% | 2.4 x 10$^{-3}$ | 7.3% |
| W600B | 89.1 | 2.6 | 0.35 | | 91.5 | | 2.0 x 10$^{-3}$ | |
| W600C | 88.7 | 2.6 | 0.35 | | 89.4 | | 2.3 x 10$^{-3}$ | |
| W700A | 91.4 | 1.7 | 0.22 | 2.1% | 98.4 | 0.3% | 4.2 x 10$^{1}$ | 7.5% |
| W700B | 91.2 | 1.8 | 0.23 | | 98.6 | | 3.6 x 10$^{1}$ | |
| W700C | 91.5 | 1.7 | 0.22 | | 99.0 | | 3.6 x 10$^{1}$ | |
| S600A | 68.7 | 2 | 0.35 | 1.4% | 88.5 | 0.8% | 7.6 x 10$^{-2}$ | 22.7% |
| S600B | 70.5 | 2 | 0.34 | | 87.8 | | 1.1 x 10$^{-1}$* | |
| S600C | 70.8 | 2 | 0.34 | | 86.7 | | 7.0 x 10$^{-2}$ | |
| S700A | 71.2 | 1.4 | 0.24 | 3.5% | 95.5 | 0.8% | 1.2 x 10$^{2}$ | 12.0% |
| S700B | 71.5 | 1.3 | 0.22 | | 96.2 | | 9.4 x 10$^{1}$ | |
| S700C | 71.4 | 1.4 | 0.23 | | 94.4 | | 1.3 x 10$^{2}$ | |

Properties of biochars produced in triplicates (A, B, C): content of total carbon (TC), hydrogen (H), H/C molar ratio, BC$_{HyPy}$ as part of total carbon (TC) and solid-state electric conductivity (SEC). Biochars were produced from wood (W) and straw (S) pellets at 600 °C and 700 °C as indicated in the sample name. The biochars whose properties are presented here were produced independently of the biochars described in Fig 1 ("third batch" as detailed in section 2.1). Variation coefficient (Var. Coeff.) is the quotient of the standard deviation and the mean value. *: SEC of sample S600B was determined in triplicates and the average is displayed (individual measurements: 0.11764, 0.11180, 0.11368 mS cm$^{-1}$, Var Coeff = 2.1%)

of saturation. For a better mechanistic understanding, future studies should include even higher pyrolysis temperatures and the measurement of reference materials such as conductive carbon black and defined nano carbon species.

It was unexpected that the conductivity of straw biochar was systematically higher than that of wood biochar in the temperature range 400–700 °C, despite the higher ash content of straw. Ash content negatively impacts the SEC of biochar, as demonstrated by both artificial mixtures (data not shown) and intrinsic variations [30], where higher ash fractions consistently lead to lower SEC when biochars are produced at similar temperatures from the same feedstock. At the same time, the presence of ash-forming substances also influences the speciation of pyrogenic carbon compounds through catalytic effects [31] and could therefore lead to an increase in SEC, depending on the composition of the ash. To investigate this in greater detail, a database with more than two different biomass and ash compositions would be required, which was beyond the scope of this study.

The increase in the BC$_{HyPy}$ content with increasing pyrolysis temperature and decreasing H/C molar ratios fits into the context of the literature presented above and confirms the previous HyPy studies with lab-produced biochars [7,12,13,15]. Remarkably, the data show a saturation in BC$_{HyPy}$ content when pyrolysis temperatures exceeded 600 °C and 680 °C for wood and straw, respectively, with BC$_{HyPy}$ > 90% of TC. This is in line with the sigmoidal-like progression of BC$_{HyPy}$ observed by McBeath and colleagues in pyrolysis experiments conducted at 300–900 °C [12]. Further experiments with higher pyrolysis temperatures and/or longer residence times should investigate in more detail if BC$_{HyPy}$ plateaus > 90% TC or if (virtually) all carbon in biochar can be BC$_{HyPy}$. The latter can be expected but is not observed so far, which may indicate an artefact or contamination in the analytical HyPy.

Howell and colleagues [15] suggested a limit of 75 wt% BC$_{HyPy}$ in biochar, which was exceeded by several samples in this study, with up to 90.6 wt% BC$_{HyPy}$, representing as much as 99.0% of its total carbon content (W700C, Table 1). Interestingly, Howell and colleagues produced biochar from woody biomass at temperatures of up to 800 °C and up to 10 min holding time (+ 100 °C/min heating rate), which at first glance would appear to be comparable to the conditions used in this study, as carbon speciation and aromaticity in particular are controlled by feedstock selection, (maximum) pyrolysis

temperature, and the residence time in the pyrolysis setup [27]. While the $BC_{HyPy}$ content given as the percentage of the total biochar weight was lower in straw than in wood biochar, which is due to the higher ash content of the straw feedstock, feedstock selection had no consistent impact on the $BC_{HyPy}$ when expressed as a ratio to the TC content of the biochar.

McBeath and colleagues quantified $BC_{HyPy}$ in biochars produced at up to 900 °C from a broad range of biomasses, which covered ash content of 0.1–39.8% and suggested that higher ash content, and specifically the content of amorphous silica, may inhibit the formation of polycondensated structures and thus reduce $BC_{HyPy}$. They performed pyrolysis in batches of 20–200 g of biomass in a muffle furnace flushed with nitrogen and controlled the temperature in the biomass bed and held the desired temperature for 1 h. Their data on $BC_{HyPy}$ of biochars from pine wood and corn stover is in good agreement to the data on biochar from softwood and straw presented in this study, respectively (Fig 2a and 2b). For biochars produced at 500 °C or less, the present study showed higher $BC_{HyPy}$. In our study, the pyrolysis temperature was measured on the reactor wall. We observed selectively that the pyrolytic system no longer had to be actively heated when performing pyrolysis at 400–500 °C and that in some cases, temperatures above the set pyrolysis temperature were measured as the pyrolysis process was obviously exothermic in this temperature range. These observations were not systematically documented but are generally in line with literature [32] and measured temperatures did not deviate more than 5–10% from the set pyrolysis temperature between 400–500 °C. Still, higher $BC_{HyPy}$ content compared to McBeath *et al.* in the range of 400–500 °C might be the result of actually higher pyrolysis temperatures due to exothermal reactions.

Howell and colleagues used pulverized biomass sieved to < 0.425 mm, while considerably larger biomass pellets were used in the present study. They had a diameter of 6 mm and a length of approximately 5–10 mm (S1 Fig). The biomass particle geometry affects the heating rate and gas exchange during pyrolysis and thus impacts biochar carbon speciation, which may explain the different results [33,34]. Moreover, Howell et al. performed thermal treatment in a thermogravimetric analyzer using only 100 mg biomass under inert gas or oxygen, which reduces secondary pyrolysis reactions [15]. The latter are known to result in highly aromatized carbon species, as demonstrated in industrial pyrolysis devices [35]. The resulting biochars consistently had lower $BC_{HyPy}$ contents at similar H/C molar ratios compared to the biochars produced in our study (Fig 2c). A high (>90 wt% daf) $BC_{HyPy}$ content was only achieved by Howell et al. when some type of gasification was performed (thermal treatment under the supply of oxygen that is not sufficient for full oxidation) [15,36]. This highlights the need to conduct analyses of "real-life" industrial biochars. The use of a pilot plant in the present study was a compromise between practice-oriented, industrial-like pyrolysis conditions and the possibility of testing a range of pyrolysis temperatures under otherwise constant conditions.

## 4.2  Prospects and limits of using $BC_{HyPy}$ in multi-pool decay models

The IPCC suggested estimating biochar persistence for national greenhouse gas inventories via the pyrolysis temperature [37], which is (supposedly) simple and, above all, inexpensive. Also, our data (Fig 2a) could be interpreted in this way. However, the reality is more complex: In our experiments on PYREKA, the temperature was the single difference in pyrolysis conditions, whereas other factors impacting biochar properties, such as reactor design, particle size of biomass, residence time of solid and gaseous pyrolysis products, and residual oxygen concentration in the reactor, were constant [38]. As such, our findings to this end should not be considered universally applicable and cannot be directly extrapolated to comparisons of biochars derived from different feedstocks or subjected to varying production processes. In practice, different reactor designs and a wide range of pyrolysis conditions impact biochar properties despite the general consensus that pyrolysis temperature is the most important pyrolysis process determining biochar properties [15,21,27,33,36,39]. Moreover, determining pyrolysis temperature in practice is often challenging to impossible due to moving parts in most reactors and the challenge of establishing ideal heat transfer between thermocouples and the biomass [15,40]. Also, the heterogeneity of biochar (e.g., with regard to PAH contents) suggests considerable variability of temperature distribution within an industrial pyrolysis reactor [41]. Thus, pyrolysis temperature should not be used for the parametrization of decay models

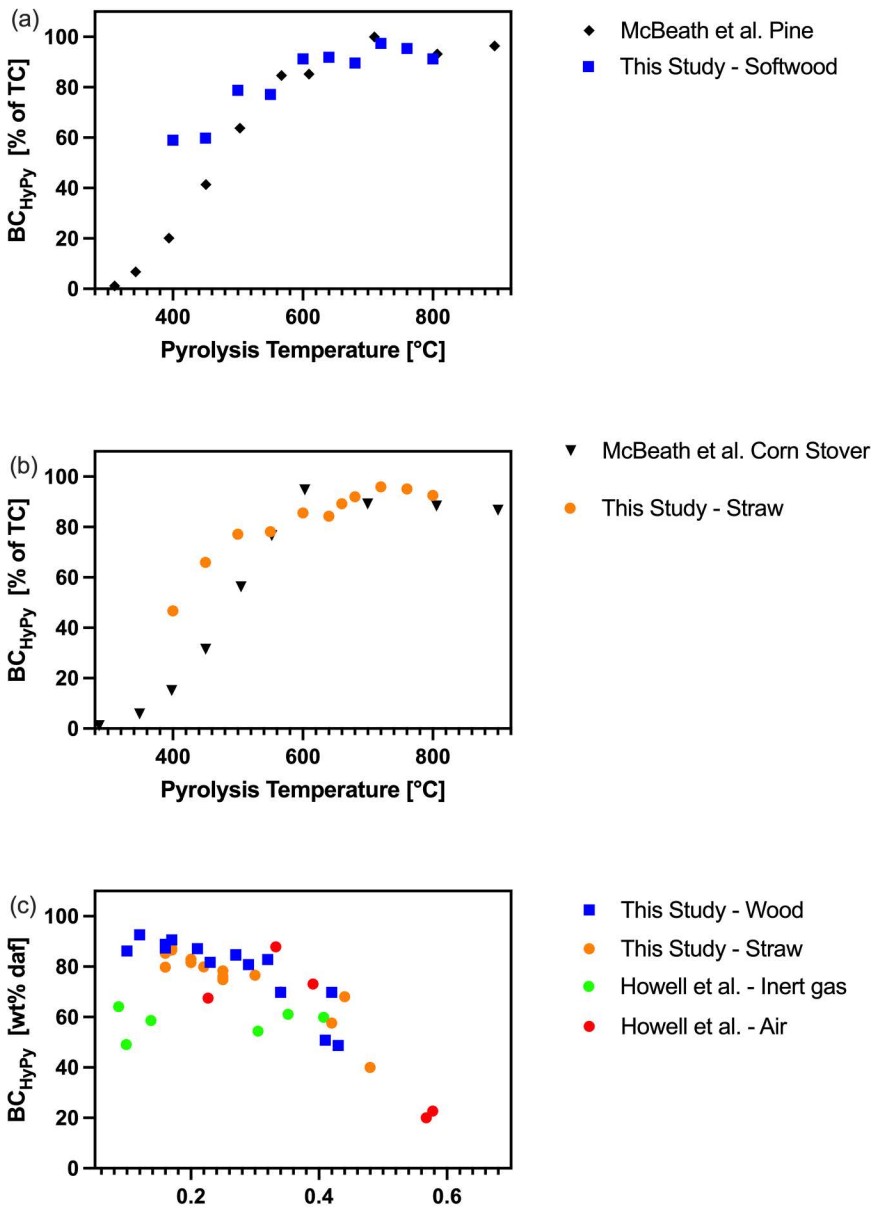

**Fig 2. Comparison of the content of BC_HyPy from biochar obtained from a continuously operating auger reactor (this study) with similar studies: McBeath et al. [12] pyrolyzed batches of 20-200 g of pre-dried pine wood (a) and corn stover (b) under nitrogen flow in a muffle furnace.** H/C molar ratio was not available in this study. Howell et al. (c) performed thermal treatment of pulverized woody biomass (<425 μm) in a thermogravimetric analyzer at 300-800 °C for 1-10 minutes under flow of air or nitrogen as an inert gas; only data points with H/C molar ratio < 0.7 were included.

for individual biochars. Instead, biochar decay models must be parameterized by analytical data of the produced biochar. Robust and, in the best case, simple methods are needed to enable high-throughput analysis of biochar persistence.

Initially, HyPy was not designed to quantify a persistent biochar carbon fraction. Instead, its purpose was to isolate black carbon from organic impurities in environmental samples. The HyPy residue is an operationally defined, thermally stable carbon fraction (H/C < 0.5, > 7-ring polycondensed clusters), which is quantified with high reproducibility and

precision (triplicate measurements typically within ±2% variation coefficient). As there is strong consensus in the literature that highly condensed aromatic clusters in biochar can be considered persistent (but not inert) [42–44], $BC_{HyPy}$ could be one factor to parameterize multi-pool decay models. Still, HyPy does not measure the actual size and speciation of these aromatic clusters but provides an operationally defined threshold measure for the degree of poly-condensation. To study the speciation of the $BC_{HyPy}$ fraction, X-ray diffraction [23,27] can be employed and used for correlations with other biochar properties.

Electric conductivity could be an indicator for the speciation of $BC_{HyPy}$. SEC was in the range of $10^1$–$10^3$ for biochars produced at 680–800 °C, which obviously differ in carbon speciation based on previous research [27], despite showing plateauing $BC_{HyPy}$ at >90% TC. To better understand the role of SEC as an indicator of biochar persistence, further investigations are necessary to determine whether and to what extent SEC could also be influenced by parameters other than carbon speciation, e.g., the ash content. Solid state electric conductivity may be one of the easiest methods in biochar persistence analysis to perform with little experimental equipment required.

Non-$BC_{HyPy}$ may include more or less alkanes, heterocyclic aromatic compounds, PAHs with up to seven aromatic rings, and their alkylated counterparts, as revealed by GC-MS of the compounds volatilized during HyPy. It may also include some larger aromatic molecules that cannot be quantified with conventional GC/MS [8]. Its composition depends on feedstock, pyrolysis temperature, and further biochar production conditions [7,9,45]. Understanding the stability and fate of non-$BC_{HyPy}$ in the environment is needed to quantify the time-dependent carbon sequestration of the less-persistent fraction of a given biochar [46]. It would allow to distinguish between the persistent, semi-persistent, and labile fractions of biochar and to derive a time-dependent carbon-sink accounting curve for the total biochar carbon applied to soil or materials.

Both SEC and $BC_{HyPy}$ correlate well with the H/C molar ratio, which is currently used to approximate a stable carbon fraction in individual biochars based on incubation-derived data on biochar persistence [2–4]. Other studies correlate biochar persistence with its O/C molar ratio [15,47]. However, the O content of biochar is usually calculated after quantifying C, H, N, S, and ash content with insufficient precision [48], while the direct measurement is not standardized yet [24]. Thus, the determination of O content in praxis is less reliable than most other biochar properties.

Another approach to identify a persistent carbon fraction in biochar is the analysis of macerals (organic minerals) in the carbonaceous material according to guidelines of the International Committee for Coal and Organic Petrology – ICCP [49–52]. Here, light microscopy is used to identify structures in an embedded and polished biochar sample. The reflectance of visible light is then determined microscopically according to ISO 7404−5 (vitrinite reflectance) to quantify the content of the maceral inertinite, which is considered the most recalcitrant maceral. However, the name inertinite does not mean that this maceral is inert, but that it is far less reactive than others [53]. Future research should compare the different persistence proxies, including elemental analysis (H/C and O/C molar ratios), HyPy, SEC, vitrinite reflectance and other methods, preferably on industrial biochars. There is an urgent need to consolidate the findings from physico-chemical characterization of biochar [47,54], from controlled incubation experiments [5,44] and from field trials [55,56] into a unified understanding of biochar persistence in the environment, as the current data and their interpretation are still perceived as contradictory.

## 5. Conclusion

Precise quantification of carbon sinks, both in terms of their size and their lifetime, is a prerequisite for a well-grounded deployment of negative emission technologies. This study suggests $BC_{HyPy}$ and SEC as novel analytical tools to improve the assessment of biochar persistence. Our results indicate that both parameters correlate strongly with the pyrolysis temperature and the H/C molar ratio of experimental biochars produced under highly controlled conditions within a temperature gradient. This supports the use of these parameters as proxies for the persistence of biochars produced in practice. $BC_{HyPy}$ quantification confirms that highly condensed aromatic structures become dominant at pyrolysis temperatures above 600 °C (for biochar from wood) and 680 °C (for biochar from straw), in agreement with previous findings on

biochar thermal stability. While pyrolysis temperature and feedstock selection remain key factors for biochar stability, our findings underscore the importance of additional process parameters, including reactor design and biomass particle size to be considered in future research. Simple biochar production parameters such as (highest treatment) temperature are not sufficient to predict biochar properties reliably.

Regarding HyPy and SEC, future studies should include pyrolysis temperatures above 800 °C and/or longer residence times to understand whether and when saturation occurs in the correlation of the H/C molar ratio, SEC, and HyPy. Solid-state electric conductivity and $BC_{HyPy}$ should be quantified in biochars used in extended incubation studies and respective non-incubation retention samples. This would require characterization of the non-$BC_{HyPy}$ fraction, e.g., by suitable GC-MS and/or $^{13}C$ nuclear magnetic resonance spectroscopy to quantify which species are actually degraded. The influence of biomass must also be investigated in more detail, especially that of biomass with higher ash contents. In the present study, straw was used, while potential feedstock such as digestate or sewage sludge has still higher ash contents.

The present study provides an impetus for the further development of multi-pool degradation models for biochar that will likely include further analytical methods. The presented results of HyPy and solid electric conductivity must now be compared and reconciled with other proposed characterization methods, in particular vitrinite reflectance.

## Supporting information

**S1 Table. Feedstock composition.** Analytical methods used and data on the composition of the biomass used for biochar production. LOQ = limit of quantification.
(PDF)

**S1 Fig. Feedstock pellets.** Wood pellets used for pyrolysis. Pens serve as a size reference. Pellets for the production of biochars at 400–600 °C (right) were more dense than the other pellets (left).
(PDF)

**S2 Table. Properties of biochar.** Content of total carbon (TC), hydrogen (H), H/C molar ratio, $BC_{HyPy}$ as part of total carbon (TC), $BC_{HyPy}$ of total biochar mass, $BC_{HyPy}$ of the dry and ash free (daf) of biochar, and solid-state electric conductivity (SEC). Biochars were produced from wood (W) and straw (S) pellets at 400–800 °C as indicated in the sample name.
(PDF)

## Author contributions

**Conceptualization:** Nikolas Hagemann, Hans-Peter Schmidt.

**Funding acquisition:** Nikolas Hagemann, Colin E. Snape.

**Investigation:** Nikolas Hagemann, Jannis Grafmueller, Silvio Vosswinkel, Clement N. Uguna.

**Methodology:** William Meredith, Colin E. Snape.

**Writing – original draft:** Nikolas Hagemann.

**Writing – review & editing:** Hans-Peter Schmidt, Thomas D. Bucheli, Jannis Grafmueller, Volker Herdegen, William Meredith, Clement N. Uguna, Colin E. Snape.

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
