## [Decision Letter · Decision Letter 0]

10 Jul 2025

Dear Dr. Hagemann,

Thank you for submitting your manuscript to PLOS ONE. After careful consideration, we feel that it has merit but does not fully meet PLOS ONE’s publication criteria as it currently stands. Therefore, we invite you to submit a revised version of the manuscript that addresses the points raised during the review process.

We look forward to receiving your revised manuscript.

Kind regards,

Ivan P. Kozyatnyk, Ph.D.

Academic Editor

PLOS ONE

Journal Requirements:

“The research at the University of Nottingham was supported by: (i) the Biotechnology and Biological Sciences Research Council [BBSRC, the Biochar Demonstrator, grant number BB/V011596/1] as part of the UKRI Greenhouse Gas Removal programme and (ii) the Department of Energy Security and Net Zero (DESNZ) through the Direct Air Capture and Greenhouse Gas Removal Programme Phases 1 and 2 for the grant "Bio-waste to Biochar (B to B) via Hydrothermal Carbonisation and Post-Carbonisation”. The research at Ithaka was funded by the Deutsche Forschungsgemeinschaft (DFG, German Research Foundation) – 467391808. “

“I have read the journal's policy and the authors of this manuscript have the following competing interests: Hans-Peter Schmidt reports a relationship with Carbon Standards AG that includes: board membership. Nikolas Hagemann reports a relationship with Carbon Standards AG that includes: board

membership. All other authors have declared that no competing interests exist.”

4. Please include a copy of Table 2 which you refer to in your text on page 10.

Reviewers' comments:

Reviewer's Responses to Questions

**Comments to the Author**

1. Is the manuscript technically sound, and do the data support the conclusions?

Reviewer #1: Yes

Reviewer #2: Yes

Reviewer #3: Yes

Reviewer #4: Yes

2. Has the statistical analysis been performed appropriately and rigorously?

Reviewer #1: Yes

Reviewer #2: Yes

Reviewer #3: Yes

Reviewer #4: No

3. Have the authors made all data underlying the findings in their manuscript fully available?

Reviewer #1: Yes

Reviewer #2: Yes

Reviewer #3: Yes

Reviewer #4: No

4. Is the manuscript presented in an intelligible fashion and written in standard English?

Reviewer #1: Yes

Reviewer #2: Yes

Reviewer #3: Yes

Reviewer #4: Yes

Reviewer #1: Dear authors,

The article “Proxies for use in Biochar Decay Models: Hydropyrolysis, Electric Conductivity, and H/Corg molar ratio” proposes the suitability of hydropyrolysis, solid state electric conductivity for the stability of biochars and their correlation with pyrolysis temperature and H/Corg. Biochars were produced in a temperature range of 400-800°C in a continuously operated pilot-scale auger reactor. The overall aim is to evaluate suitability of hydropyrolysis and solid state electric conductivity for the stability of biochars and their correlation with pyrolysis temperature and H/Corg for the parametrization of novel decay model for industrial prepared biochars to pyrogenic carbon capture and storage, including prospects and limits of using BChyPy in multi-pool decay models. A comparison with other approaches to identify a persistent fraction in biochar was made and discussed, including future research topics.

The paper is well-written and documented with up-to-date references.

However, there are some issues which require more explanation.

A list of minor comments is addressed to the authors.

Minor comments

1. Line 25: Abbreviation “PYCSS” (pyrogenic carbon capture and storage) should be explained when first introduced.

2. Line 39: “Hydropyrolysis” instead of “Hydropyrolyis”. Please comment.

3. Line 165: “Fig. 1c” instead of “Fig 1C”

4. Line 176: What is the reason for the high variation coefficients for SEC measurements in contrast to the much lower variation coefficients for BCHyPy and H/C molar ratios? Please comment.

5. Line 186, Table 1: Why different units were used for SEC values in Table 1 (µS cm-1) and Fig.1a, 1b and 1f (mS cm-1)? Please comment.

6. Line 225: “Table 1” instead of “Table 2”. Please comment.

7. Lines 366-367: “to quantify which special are actually degraded”: Not clear to me. Please comment.

Reviewer #2: The manuscript presents a well-designed and executed study on the use of physicochemical proxies, hydropyrolysis (HyPy), solid-state electric conductivity (SEC), and H/C molar ratio, to predict biochar persistence. The experimental approach is rigorous, employing a pilot-scale auger reactor to produce biochars from straw and wood across a temperature gradient (400–800°C) under controlled conditions. Characterization methods, including elemental analysis, SEC, and HyPy, are well-documented and supported by triplicate samples, ensuring reproducibility. The results convincingly demonstrate strong correlations between BCHyPy, SEC, pyrolysis temperature, and H/C ratios, reinforcing their utility in decay models. Notably, the plateau in BCHyPy content above 600–680°C aligns with existing literature, while the unexpected higher SEC in straw biochars, despite their lower carbon content, suggests additional factors such as ash composition may influence conductivity, a point that could be explored further in the discussion.

The discussion effectively contextualizes these findings within the broader literature, particularly through comparisons with prior studies by McBeath et al. and Howell et al. While the authors acknowledge limitations (e.g., potential variability in pyrolysis temperature measurement), a deeper exploration of mechanisms behind SEC differences, such as linking straw biochars’ higher conductivity to their ash mineralogy (e.g., potassium or silica content), would strengthen the narrative. Additionally, a brief rationale for prioritizing H/C over O/C ratios, despite the latter’s historical use in persistence models, would provide a more balanced perspective.

To enhance the manuscript, error bars in Figure 1 would better reflect the variability observed in triplicate measurements, especially for SEC (7–23% variation). The figures and tables otherwise clearly support the conclusions, and the literature review is thorough, though tighter integration of recent debates (e.g., O/C ratios) could further solidify the proxies’ rationale.

The Supporting Information (SI) complements the main text but could be refined for greater impact. Table S1’s detailed biomass composition (ash, trace metals) is valuable for understanding feedstock variability, though grouping elements by functional relevance (e.g., macronutrients vs. ash-forming minerals) and noting their potential influence on conductivity (e.g., high potassium in straw) would streamline interpretation. Table S2’s systematic listing of biochar properties across temperatures and pressures is robust, but clarifying whether statistical analysis was applied to batch replicates would reinforce reproducibility. Figure S1, while visually confirming pellet consistency, feels peripheral unless pellet density’s role in heat transfer is explicitly tied to pyrolysis outcomes.

Minor adjustments would further polish the SI: correcting chemical notation (e.g., "Al₂O₃" instead of "Al2O3") and cross-referencing key SI results in the main discussion (e.g., linking straw’s high sulfur ash to SEC trends) would tighten integration.

In summary, the study makes a significant contribution by proposing SEC and BCHyPy as reliable proxies for biochar persistence, with clear implications for carbon sequestration strategies. The manuscript is scientifically sound and merits publication after minor revisions, such as refining the discussion of SEC mechanisms, adding error bars to figures, and streamlining the SI. With its technical rigor and logical flow, this work will be a valuable addition to the literature on biochar stability and decay modeling.

Reviewer #3: The presented by Hagemann et al. manuscript titled "Proxies for use in Biochar Decay Models: Hydropyrolysis, Electric Conductivity, and H/Corg molar ratio" is very interesting and well written. My few comments/questions are in the attatched pdf file.

Reviewer #4: I read the manuscript «Proxies for use in Biochar Decay Models: Hydropyrolysis, Electric Conductivity, and H/Corg molar ratio,» by Hagemann et al. [PONE-D-25-30085]. The work explores the characterization of biochar through two techniques, namely hydropyrolysis where the biochar is treated with H2 at around 550°C in the presence of a catalyst to determine the non-converted fraction; and solid-state conductivity measurements. Both these measures are suggested as robust characteristics for biochar that could be used for calibrating bio-degradation models. The work presents a comprehensive set of experimental data on biochar produced in a continuous bench scale pyrolizer from two different feedstock at different temperatures.

The manuscript is well written, and the presented data is of high quality. The discussion presented at the end of the paper is on a high level and proves to be of value to a broader audience. I recommend publication of the work after few minor clarifications and additions to the manuscript.

Minor comments

1. Line 121 on page 5 mentions a residence time of 10 min the pyrolyzer. How was this residence time determined? Does it depend on the feed rate?

2. Line 141, page 6 describes the sample preparation for the HyPy measurements. The description, however, leaves open some gaps. Specifically, what was the methanol fraction in the water/methanol mixture to disperse the Mo catalyst? Also, was the biochar sample tried after being impregnated with the Mo-catalyst solution?

3. Table 1, page 8: Which columns shows content of total carbon? Is this the second column with the header “C (%)”?

4. Page 11, line 274: “It would be very appealing …” I do not quite understand the statements here, especially when contrasting them to the passage in the middle of page 12, i.e. line 291 “the pyrolysis temperature should not be used for the parametrization of decay models”. I suggest rephrasing the passage in line 274-277.

5. I found the note on the temperature overshoot and exothermic pyrolysis reaction on page 10, line 244 most interesting. As a remark, experimental evidence for exothermic pyrolysis reactions was first reported by Park, Atreya and Baum, Combustion and Flame 157 (2010) 481-494. Maybe the authors find this useful, or maybe not.

Typos:

* Page 4, line 88: “BCHyPy” requires subscript format

Remark: Can the data shown in Fig. 1 and Fig. 2 also be provided as a table in, e.g. supplementary material

**Do you want your identity to be public for this peer review?** For information about this choice, including consent withdrawal, please see our Privacy Policy

Reviewer #1: No

Reviewer #2: No

Reviewer #3: No

Reviewer #4: **Yes: ** Matthaus Babler

---

## [Decision Letter · Decision Letter 1]

29 Jul 2025

Proxies for use in Biochar Decay Models: Hydropyrolysis, Electric Conductivity, and H/Corg molar ratio

PONE-D-25-30085R1

Dear Dr. Hagemann,

We’re pleased to inform you that your manuscript has been judged scientifically suitable for publication and will be formally accepted for publication once it meets all outstanding technical requirements.

Kind regards,

Ivan P. Kozyatnyk, Ph.D.

Academic Editor

PLOS ONE

Additional Editor Comments (optional):

Reviewers' comments:

Reviewer's Responses to Questions

**Comments to the Author**

Reviewer #1: All comments have been addressed

Reviewer #2: (No Response)

Reviewer #3: All comments have been addressed

Reviewer #4: All comments have been addressed

2. Is the manuscript technically sound, and do the data support the conclusions?

Reviewer #1: Yes

Reviewer #2: Yes

Reviewer #3: Yes

Reviewer #4: Yes

3. Has the statistical analysis been performed appropriately and rigorously?

Reviewer #1: Yes

Reviewer #2: Yes

Reviewer #3: Yes

Reviewer #4: Yes

4. Have the authors made all data underlying the findings in their manuscript fully available?

Reviewer #1: Yes

Reviewer #2: Yes

Reviewer #3: Yes

Reviewer #4: Yes

5. Is the manuscript presented in an intelligible fashion and written in standard English?

Reviewer #1: Yes

Reviewer #2: Yes

Reviewer #3: Yes

Reviewer #4: Yes

Reviewer #1: I have read the revised article “Proxies for use in Biochar Decay Models: Hydropyrolysis, Electric Conductivity, and H/Corg molar ratio”. Authors have addressed the comments and questions raised by the reviewers in a detailed way and have implemented additional information according to the suggestions of the reviewers.

Minor comment

Lines 123-124: “in 50°C steps, second batch in 20°C steps” instead of “in 50° steps, second batch in 20° steps”

Reviewer #2: I want to thank the authors for their thoughtful and transparent responses to the reviewer feedback. The revised manuscript integrates the suggested changes in a consistent and scientifically sound manner.

In particular, the expanded discussion on the unexpectedly high SEC values in straw-derived biochars stands out. The authors now consider plausible explanations, such as ash content and potential catalytic influences, which adds welcome nuance to their interpretation. They've also done well to acknowledge the limitations of the current dataset with appropriate candor.

Their decision to prioritize H/C over O/C molar ratios as a proxy for biochar persistence is clearly justified, balancing analytical constraints with methodological reasoning in a way that feels both practical and well-supported.

Supporting Information is now more user-friendly, with clearer links to the main text and cleaner formatting overall. Minor issues with figure captions and notation have been addressed effectively.

Regarding Figure 1, although error bars weren't added, the authors have clarified that the data points in question weren't replicated and have made this explicit in the manuscript. That explanation is reasonable and does not detract from the overall quality of the work.

Beyond the points raised in my own review, I’ve also read the feedback from the other reviewers. The authors have addressed those comments with similar diligence, thereby providing clarifications on methodological details (such as residence time in the pyrolyzer, SEC units, and catalyst loading), feedstock selection, and the underlying experimental rationale. They’ve also corrected various minor textual and formatting issues flagged by others.

All things considered, the revised manuscript is scientifically robust, clearly written, and thoroughly responsive to the feedback provided. I see no outstanding concerns and recommend it for acceptance.

Reviewer #3: The authors, Hagemann et al., have revised the manuscript addressing all reviwers' comments and questions. Now I recommend the manuscript titled "Proxies for use in Biochar Decay Models: Hydropyrolysis, Electric Conductivity, and

H/Corg molar ratio" to be published in PLOS One Journal.

Reviewer #4: All comments have been addressed and the manuscript has been modified accordingly. I congratulate the authors to their work.

**Do you want your identity to be public for this peer review?** For information about this choice, including consent withdrawal, please see our Privacy Policy

Reviewer #1: No

Reviewer #2: No

Reviewer #3: No

Reviewer #4: No

---

## [Editor Report · Acceptance letter]

PONE-D-25-30085R1

PLOS ONE

Dear Dr. Hagemann,

I'm pleased to inform you that your manuscript has been deemed suitable for publication in PLOS ONE. Congratulations! Your manuscript is now being handed over to our production team.

Kind regards,

on behalf of

Dr. Ivan P. Kozyatnyk

Academic Editor

PLOS ONE